# The role of traditional Chinese medicine on fracture surgery, hospitalization, and total mortality risks in diabetic patients with osteoporosis

Yi-Chen Liu[1], Chi-Hsiang Chung[2,3], Chien-Jung Lin[4], Sheng-Chiang Su[1], Feng-Chih Kuo[1], Jhih-Syuan Liu[1], Peng-Fei Li[1], Chia-Luen Huang[1], Li-Ju Ho[1], Chun-Yung Chang[5], Ming-Shiun Lin[1], Chih-Ping Lin[1], An-Che Cheng[1], Chien-Hsing Lee[1], Chang-Hsun Hsieh[1], Yi-Jen Hung[1], Hsin-Ya Liu[6], Chieh-Hua Lu[1]*, Wu-Chien Chien[2,3,7]*

1 Department of Internal Medicine, Division of Endocrinology and Metabolism, Tri-Service General Hospital, School of Medicine, National Defense Medical Center, Taipei, Taiwan, ROC, 2 School of Public Health, National Defense Medical Center, Taipei, Taiwan, ROC, 3 Taiwanese Injury Prevention and Safety Promotion Association, Taipei, Taiwan, ROC, 4 Department of Chinese Medicine, Tri-Service General Hospital, National Defense Medical Center, Taipei, Taiwan, ROC, 5 Department of Internal Medicine, Division of Endocrinology and Metabolism, Kaohsiung Armed Forces General Hospital, Kaohsiung, Taiwan, ROC, 6 BeYoung Research Institute, Taipei, Taiwan, ROC, 7 Department of Medical Research, Tri-Service General Hospital, National Defense Medical Center, Taipei, Taiwan, ROC

* undeca2001@gmail.com (C-HL); chienwu@mail.ndmctsgh.edu.tw (W-CC)

**Data Availability Statement:** Information is accessible through the Taiwan National Health Insurance (NHI) Bureau's National Health

## Abstract

### Background

Studies have confirmed that osteoporosis has been considered as one of the complications of diabetes, and the health hazards to patients are more obvious. This study is mainly based on the Taiwan National Health Insurance Database (TNHID). Through the analysis of TNHID, it is shown that the combined treatment of traditional Chinese medicine (TCM) medicine in patients of diabetes with osteoporosis (T2DOP) with lower related risks.

### Methods

According to the study design, 3131 patients selected from TNHID who received TCM treatment were matched by 1-fold propensity score according to gender, age, and inclusion date as the control group. Cox proportional hazards analyzes were performed to compare fracture surgery, hospitalization, and all-cause mortality during a mean follow-up from 2000 to 2015.

### Results

A total of 1055/1469/715 subjects (16.85%/23.46%/11.42%) had fracture surgery/inpatient/all-cause mortality of which 433/624/318 (13.83%/19.93%/10.16%) were in the TCM group) and 622/845/397 (19.87%/26.99%/12.68%) in the control group. Cox proportional hazards regression analysis showed that subjects in the TCM group had lower rates of fracture surgery, inpatient and all-cause mortality (adjusted HR = 0.467; 95% CI = 0.225–0.680,

Insurance Research Database (NHIRD). However, adherence to the "Personal Information Protection Act" legally enforced by the Taiwanese government prohibits the public sharing of this data. For those interested in obtaining the data, the appropriate procedure involves submitting a formal proposal to the NHIRD via their website: http://nhird.nhri.org. tw. Researchers who are interested must possess valid Institutional Review Board documentation and must submit an application to the NHIRD. After undergoing a review process, there is a fee associated with obtaining database access rights. This ensures compliance with regulatory and ethical standards while facilitating the acquisition of valuable data for research purposes.

**Funding:** This work was supported by research grants from the Tri-Service General Hospital, No TSGH-B-112020, No TSGH-E-112258 and No 801GB112258 and the Cheng Hsin General Hospital, No:CHNDMC-112-10. The sponsor has no role in study design, data collection and interpretation, decision to publish, or preparation of the manuscript.

**Competing interests:** The authors have declared that no competing interests exist.

P<0.001; adjusted HR = 0.556; 95% CI = 0.330–0.751, P<0.001; adjusted HR = 0.704; 95% CI = 0.476–0.923, P = 0.012). Kaplan-Meier analysis showed that the cumulative risk of fracture surgery, inpatient and all-cause mortality was significantly different between the case and control groups (all log-rank p<0.001).

## Conclusion

This study provides longitudinal evidence through a cohort study of the value of integrated TCM for T2DOP. More research is needed to fully understand the clinical significance of these results.

## Introduction

Aging societies seeing rising rates of diabetes and osteoporosis each year can pose significant health problems for a nation's society [1]. The global prevalence of diabetes among people aged 20–79 years is estimated to be 10.5% (536.6 million) in 2021, rising to 12.2% (783.2 million) by 2045 [2]. Osteoporosis is considered an important contributor to fractures of the spine, hip, distal forearm, and proximal humerus, which almost always result in hospitalized loss of function [3]. The classification of bone mineral density and the diagnosis of osteoporosis was established by the World Health Organization in 1994, as measured by the widely used dual-energy X-ray absorptiometry, and a T-score equal to or less than -2.5 correlates with bone mass [4]. Osteoporosis is a recognized comorbidity in patients with diabetes [5], and an important independent risk factor for osteoporosis [6].

According to the 2019 Taiwan Diabetes Yearbook, there are more than 2.2 million people with diabetes in Taiwan, and the rate of diabetes among people > 65 years of age is as high as 25% [7]. What is worrying is that, for every four people > 65 years of age, 1% have OP, while the fracture risk is increased in patients with diabetes, who are more likely to develop OP than the general population [8]. If diabetes is added as a risk factor, the probability of fractures due to osteoporosis increases further.

In Taiwan, traditional Chinese medicine (TCM) is becoming increasingly popular as an adjunctive treatment for chronic diseases, particularly diabetes [9]. Many studies to date described treating diabetes with TCM and concluded that it can improve blood sugar to a certain extent [10]. However, it is unknown whether the severity of osteoporosis of patients with type 2 diabetes and osteoporosis (T2DOP) who receive TCM treatment will decrease in terms of reduced fracture surgery, hospitalization, or mortality rates. This study aimed to analyze data from the Taiwan National Health Insurance Research Database (NHIRD) to investigate whether T2DOP plus TCM treatment could reduce the risks of fracture surgery, hospitalization, and all-cause mortality among patients with T2DOP.

## Materials and methods

### Data sources

Patients with diabetes mellitus and osteoporosis were recruited from the Taiwan outpatient Longitudinal Health Insurance Database (LHID). We used data from NHIRD to investigate whether TCM treatment could reduce fracture surgery, hospitalization, or mortality in diabetic patients with osteoporosis over a 15-year period (2000–2015). In 1995, Taiwan launched the National Health Insurance (NHI) program and it has contracted with 97% of medical

providers in Taiwan, which has a population of approximately 23 million, or 99% of the total population of Taiwan, as of June 2009 [11]. NHIRD uses the International Classification of Diseases, Ninth Revision Clinical Modification (ICD-9-CM) to record diagnoses. All diagnoses of type 2 diabetes mellitus (T2D) and osteoporosis were made by specialist certified medical professionals, while treatment included the herbal formulae illustrated in S1 Table. The NHI randomly reviewed records for every 100 outpatient visits and every 20 inpatient claims to verify diagnostic accuracy [12].

## Study design and sample participants

Our study used a retrospective paired cohort design. From January 1, 2000 to December 31, 2015, the diagnoses of T2D and osteoporosis were selected according to the codes ICD-9-CM 250.XX (T2D) and ICD-9-CM 733.0 (osteoporosis), respectively. According to these ICD-9-CM codes, each enrolled patient had at least 3 outpatient visits during the study period, and patients who received less than 3 TCM treatments and were younger than 18 years old were excluded. Covariates included Chalson Comorbidity Index (CCI) minus T2D, level of care, gender, and age. CCI represents "comorbidity" and catastrophic illness represents "severity".

We initially included 29,166 T2DOP; however, 5,136 patients were ultimately excluded because they received TCM treatment before 2000, had no follow-up records, were younger than 18 years old, or because their gender could not be identified. Finally, 24,030 T2DOP were included in the analysis, among whom 4,155 who received TCM treatment, 19,875 who did not receive TCM treatment, and 1024 outpatient follow-up patients who received TCM treatment less than 3 times were excluded. The remaining 3131 patients who received TCM treatment were divided using 1-fold propensity score matching by gender, age, and inclusion date as a control group without TCM treatment, as shown in Fig 1.

## Outcome measures

We primarily track all study participants when undergo surgery for fractures, inpatient, or mortality from any cause. Based on the NHI program's follow-up period to the end of 2015 as the outcome measure, we will see if combining it with TCM treatment can reduce the incidence of these events.

## Statistical analysis

All statistical analyzes were performed using SPSS software version 22 (SPSS Inc., Chicago, IL, USA), R software version 4.3.1 (R Software Inc., San Francisco, CA, USA), and STATA software version 9 (StataCorp LLC., College Station, Texas, USA). Chi-square and t-tests were used to assess the distribution of categorical and continuous variables, respectively. Multivariate Cox proportional hazards regression analysis with a mixed effects model was used to determine the risk of fracture surgery, hospitalization, or death in T2DOP treated with TCM. We applied Schoenfeld's global test to test the proportional hazards assumption in the Cox proportional hazards model. Cox proportional hazards model assumes that the hazards of the different strata formed by the levels of the covariates are proportional. Schoenfeld's global test is used to test the residuals of Cox proportional model. If $P > 0.05$, it means that the proportional hazard assumption is not violated; in other words, the Cox proportional hazards model can be used [13,14]. Results of statistical analyzes are presented as hazard ratios (HR) and 95% confidence intervals (CI). Differences in the risk of fracture surgery, hospitalization, or death between groups receiving and not receiving TCM were estimated using the Kaplan-Meier method and the log-rank test. Statistical significance was determined using a two-tailed test with a p-value less than 0.05.

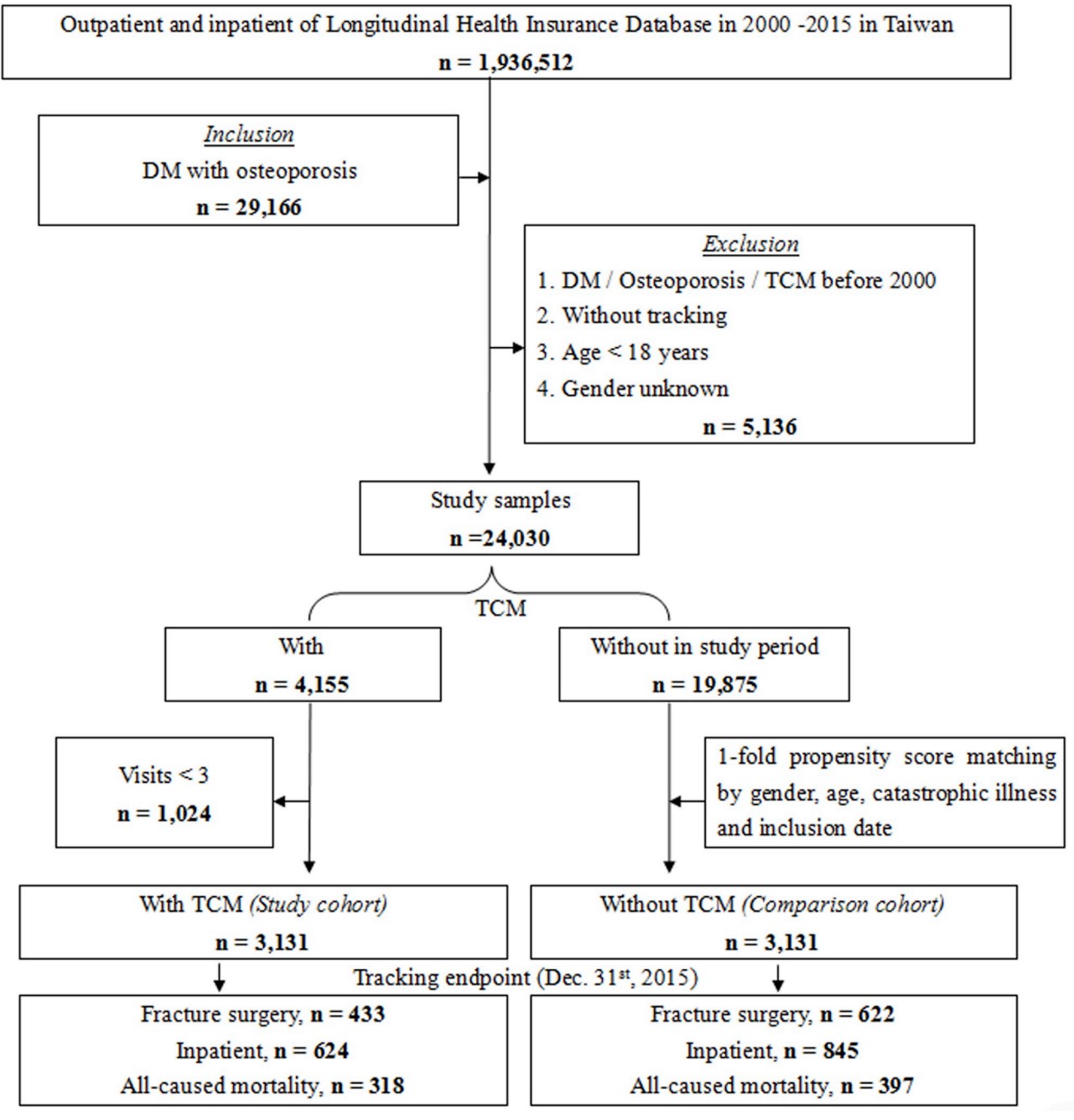

**Fig 1. The flowchart of study.**

## Ethics approval and consent to participate

Our research was performed in accordance with the World Medical Association Code of Ethics (Declaration of Helsinki). The Institutional Review Board of the Tri-Service General Hospital (TSGH) approved our study and waived the need for individual written informed consent (TSGHIRB No. E202316013).

## Results

We included 29,166 T2DOP, and excluded 5,136 patients who received TCM treatment before 2000, those who had no follow-up records, those who were younger than 18 years old, and whose gender could not be identified, finally included 24030 T2DOP. Among them, there were 4155 patients who received TCM treatment, 19875 patients who did not receive TCM treatment, and 1024 outpatient follow-ups who received TCM treatment were excluded for less than 3 times. The remaining 3131 patients who received TCM treatment were divided by 1-fold propensity score matching by gender, age, and inclusion date as control group without TCM treatment. Among T2DOP who received TCM treatment, 433 fractures underwent surgery, 624 were hospitalized, and 318 had all-caused mortality. Among T2DOP who did not receive TCM treatment, 622 fractures underwent surgery, 845 were hospitalized, and 397 had all-caused mortality, as shown in Fig 1. T2DOP who received TCM treatment through Kaplan-Meier analysis had a lower cumulative risk of fracture surgery, inpatient, and all-caused mortality than those who did not receive TCM that show in Figs 2–4 (all log-rank p<0.001).

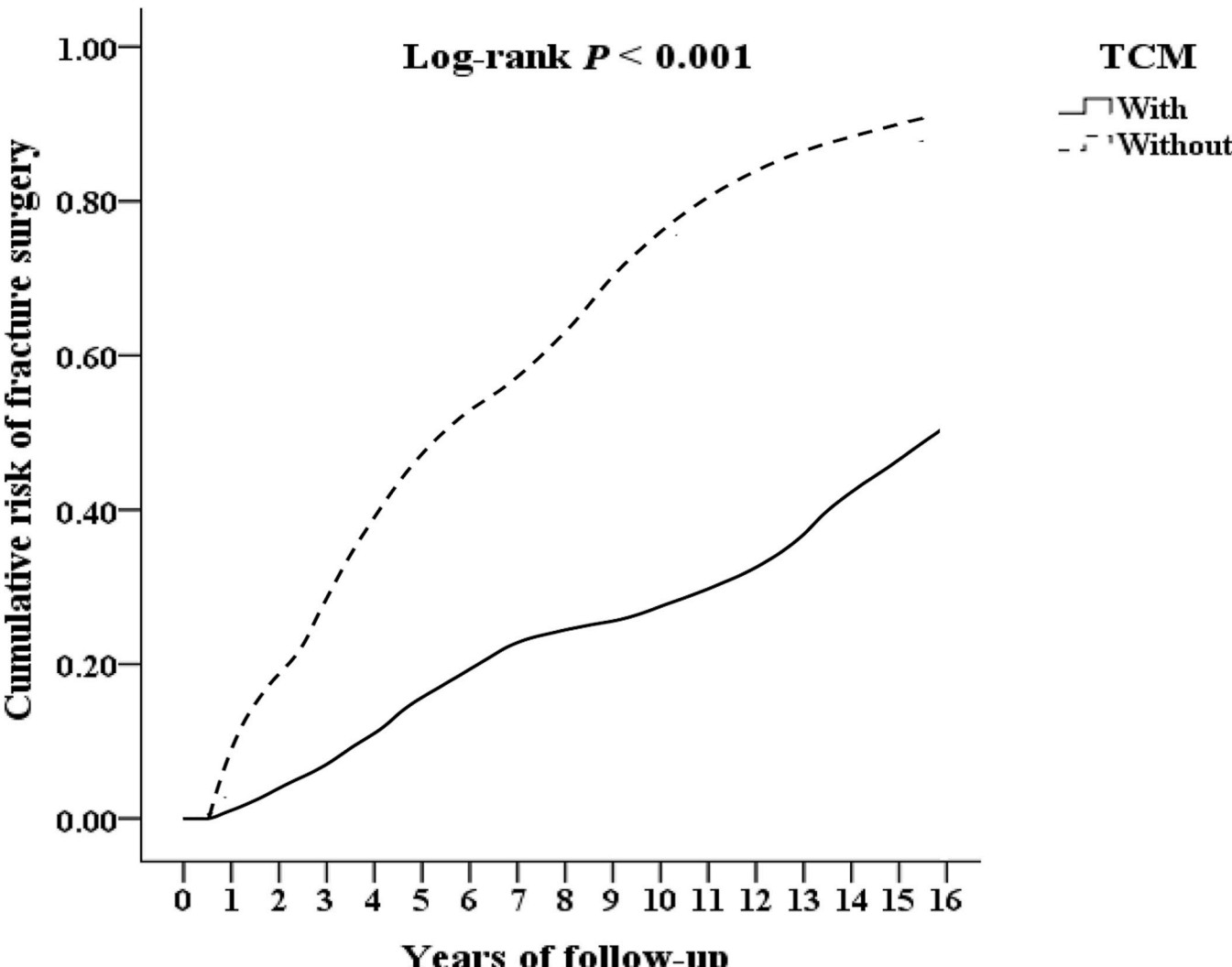

**Fig 2. Kaplan-Meier for cumulative risk of fracture surgery among patients of diabetes with osteoporosis aged 18 and over stratified by TCM with log-rank test.**

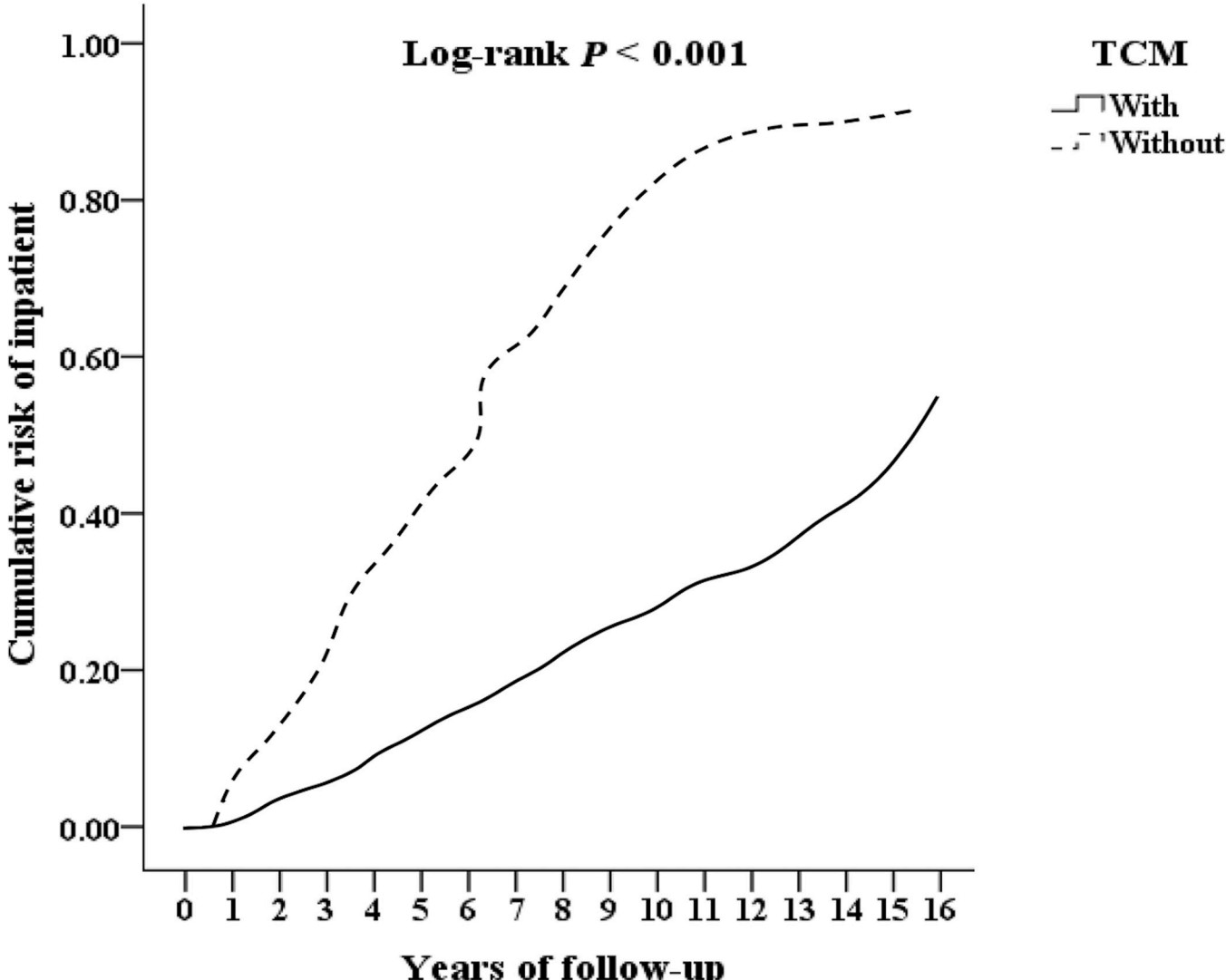

**Fig 3. Kaplan-Meier for cumulative risk of inpatient among patients of diabetes with osteoporosis aged 18 and over stratified by TCM with log-rank test.**

Baseline characteristics of the study included sex, age, comorbidities, and disease severity (Table 1). Among the 6262 T2DOP, 4212 patients (67.26%) were female, 1025 patients (32.74%) were males, and the average age of income cases was 63.50 ± 19.87 years old which 558 patients (8.91%) were younger than 50 years old, 1284 patients (20.50%) were 50–59 years old, and 4420 patients were over 60 years old (70.58%). There were no significant differences between the TCM group and the control group with respect to sex, age, catastrophic disease, revision of the Charlson comorbidity index (CCI), and covariates.

In Table 2, the fracture surgery rate, inpatient rate and all-cause mortality rate in the TCM group were lower than those in the control group at the end of follow-up. There were 1055 subjects (16.85%) underwent surgery due to fractures, 433 patients in the TCM treatment group and 622 patients in the control group (13.83% vs 19.87%, p<0.001); 1469 patients (23.46%) subjects were hospitalized, 624 cases in the TCM group, 845 patients in the control group (19.93% vs 26.99%, p<0.001); 715 patients (11.42%) died, 318 patients in the TCM group, 397 patients in the control group (10.16% vs 12.68%, p = 0.002). At the end of follow-

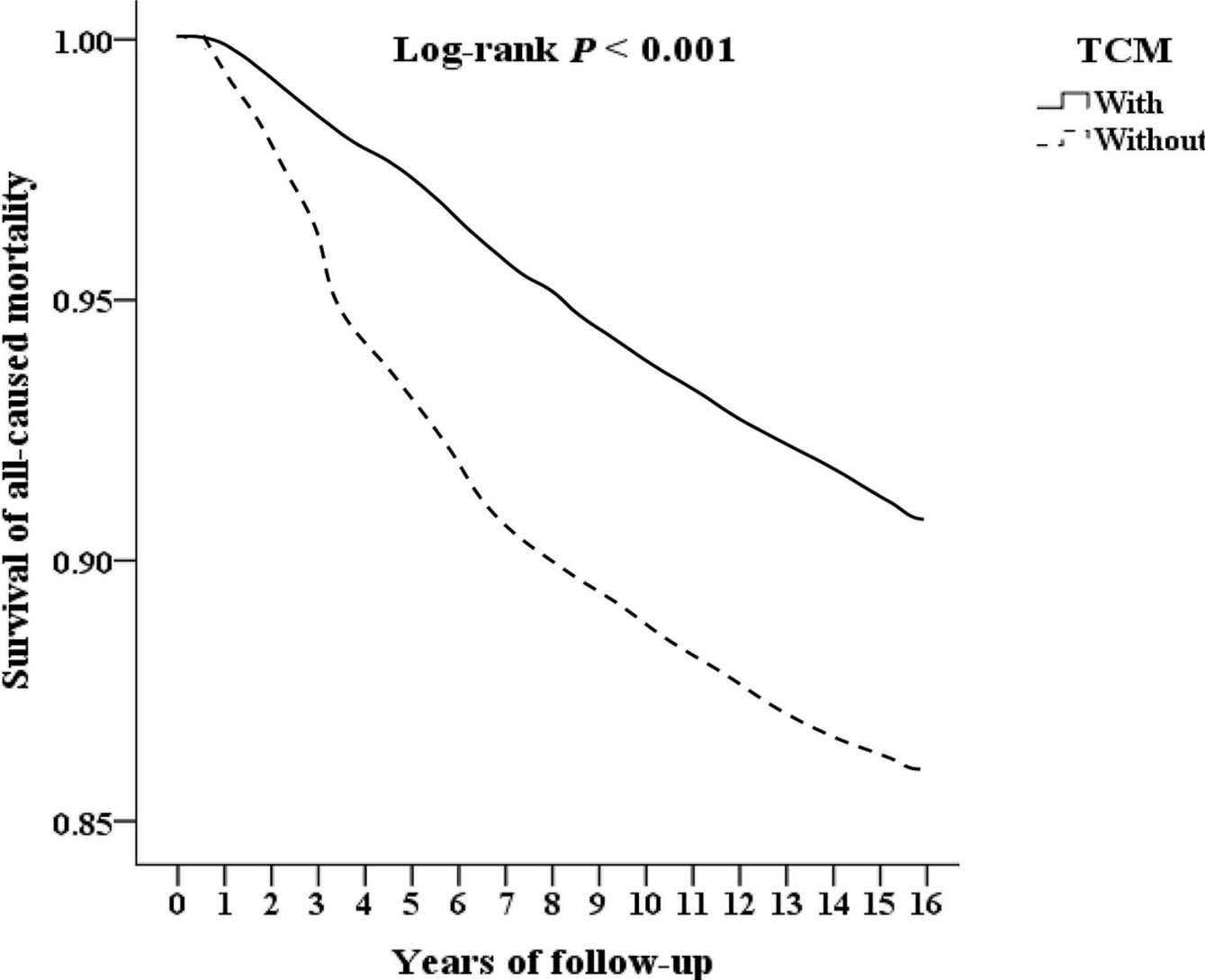

**Fig 4. Kaplan-Meier for survival of all-caused mortality among patients of diabetes with osteoporosis aged 18 and over stratified by TCM with log-rank test.**

up, there were no significant differences between the TCM group and the control group with respect to sex, age, catastrophic disease, revision of the CCI, and covariates. T2DOP who will receive TCM treatment have a higher proportion in the medical center (p<0.001).

Table 3 shows the factors that affect the Cox regression of fractures received surgical treatment, Inpatient and all-caused mortality. In patients who received surgical treatment for fractures, we saw that T2DOP who received TCM had an adjusted HR of 0.467 (95 CI = 0.225–0.680, P<0.001), and a lower proportion received surgery for fractures treat. In male patients, the adjusted HR was 1.239 (95 CI = 0.931–1.625, P = 0.067) without difference with female. The patients were 50–59 years old, the adjusted HR was 1.666 (95 CI = 1.112–2.304, P<0.001), the older patients were > = 60, the adjusted HR was 1.801 (95 CI = 1.144–2.410, P<0.001), and the patients with catastrophic illness had an adjusted HR of 1.898 (95 CI = 1.456–2.591, P<0.001), those with CCI_R adjusted HR 1.331 (95 CI = 1.193–1.618, P<0.001) had a higher proportion of fractures received surgical treatment. In hospitalized patients, we found that

**Table 1. Characteristics of study in the baseline.**

| TCM | Total | | With | | Without | | P |
|---|---|---|---|---|---|---|---|
| **Variables** | **n** | **%** | **n** | **%** | **n** | **%** | |
| **Total** | 6,262 | | 3,131 | 50.00 | 3,131 | 50.00 | |
| **Gender** | | | | | | | 0.999 |
| Male | 2,050 | 32.74 | 1,025 | 32.74 | 1,025 | 32.74 | |
| Female | 4,212 | 67.26 | 2,106 | 67.26 | 2,106 | 67.26 | |
| **Age (years)** | 63.50 ± 19.87 | | 63.44 ± 19.84 | | 63.56 ± 19.90 | | 0.811 |
| **Age groups (yrs)** | | | | | | | 0.999 |
| 18–49 | 558 | 8.91 | 279 | 8.91 | 279 | 8.91 | |
| 50–59 | 1,284 | 20.50 | 642 | 20.50 | 642 | 20.50 | |
| ≧ 60 | 4,420 | 70.58 | 2,210 | 70.58 | 2,210 | 70.58 | |
| **Catastrophic illness** | | | | | | | 0.999 |
| Without | 4,934 | 78.79 | 2,467 | 78.79 | 2,467 | 78.79 | |
| With | 1,328 | 21.21 | 664 | 21.21 | 664 | 21.21 | |
| **CCI_R** | 1.02 ± 1.13 | | 1.02 ± 1.14 | | 1.01 ± 1.11 | | 0.725 |
| **Level of care** | | | | | | | < 0.001 |
| Medical center | 2,765 | 44.16 | 1,682 | 53.72 | 1,083 | 34.59 | |
| Regional hospital | 2,150 | 34.33 | 1,035 | 33.06 | 1,115 | 35.61 | |
| Local hospital | 1,347 | 21.51 | 414 | 13.22 | 933 | 29.80 | |

*P*: Chi-square / Fisher exact test on category variables and t-test on continue variables.

T2DOP who received TCM had an adjusted HR of 0.556 (95 CI = 0.330–0.751, P<0.001), and a lower proportion of inpatients. For male patients, adjusted HR 1.301 (95 CI = 1.026–1.510, P = 0.033), the patients were 50–59 years old, the adjusted HR was 1.276 (95 CI = 0.960–1.556, P = 0.059), older > = 60, adjusted HR 1.878 (95 CI = 1.298–2.596, P<0.001), for patients with catastrophic illness, adjusted HR 1.875 (95 CI = 1.161–2.510, P<0.001), those with CCI_R adjusted HR 1.186 (95 CI = 1.005–1.350, P = 0.047) had a higher proportion of inpatient. Similarly in all-caused mortality, T2DOP who received TCM treatment had an adjusted HR of 0.704 (95 CI = 0.476–0.923, P = 0.012), and a lower proportion of deaths. For male patients, the adjusted HR was 1.333 (95 CI = 0.975–1.816, P = 0.057), the patients were 50–59 years old, the adjusted HR was 1.454 (95 CI = 1.023–2.026, P = 0.037), for older patients > = 60, the adjusted HR was 1.880 (95 CI = 1.346–2.455, P<0.001), for patients with catastrophic illness, the adjusted HR was 2.015 (95 CI = 1.465–2.810, P<0.001), those with CCI_R adjusted HR 1.190 (95 CI = 1.087–1.324, P = 0.007) will have higher all-caused mortality.

Table 4 helps us to further illustrate that T2DOP who received TCM treatment will have lower fractures rate adjusted HR 0.467 (95 CI = 0.225–0.680, P<0.001) and lower inpatient rate adjusted HR 0.556 (95 CI = 0.330–0.751, P<0.001), no matter in gender, age, disease severity or medical center, as long as T2DOP who received TCM treatment had a lower risk of receiving surgery for fracture Treatment and inpatient rate (all p<0.001). T2DOP who received TCM treatment had lower all-caused mortality, adjusted HR 0.704 (95 CI = 0.476–0.923, P = 0.012). Regardless of gender, age, disease severity, or medical center, T2DOP who received TCM treatment had lower all-caused mortality (all p<0.05). A description of the incidence rates for the main outcomes in terms of different factors, including fracture, inpatient status, and all-cause mortality stratified by variables using Cox regression is shown in S2 Table, while the average follow-up time and other information is shown in S3 Table.

Table 5 shows the factors influencing prognosis among different TCM subgroups, as assessed using Cox regression. These factors illustrate the use of TCM in the population,

**Table 2. Characteristics of study in the endpoint.**

| TCM | Total | | With | | Without | | P |
|---|---|---|---|---|---|---|---|
| **Variables** | **n** | **%** | **n** | **%** | **n** | **%** | |
| **Total** | 6,262 | | 3,131 | 50.00 | 3,131 | 50.00 | |
| **Fracture surgery** | | | | | | | < 0.001 |
| Without | 5,207 | 83.15 | 2,698 | 86.17 | 2,509 | 80.13 | |
| With | 1,055 | 16.85 | 433 | 13.83 | 622 | 19.87 | |
| **Inpatient** | | | | | | | < 0.001 |
| Without | 4,793 | 76.54 | 2,507 | 80.07 | 2,286 | 73.01 | |
| With | 1,469 | 23.46 | 624 | 19.93 | 845 | 26.99 | |
| **All-caused mortality** | | | | | | | 0.002 |
| Without | 5,547 | 88.58 | 2,813 | 89.84 | 2,734 | 87.32 | |
| With | 715 | 11.42 | 318 | 10.16 | 397 | 12.68 | |
| **Gender** | | | | | | | 0.990 |
| Male | 2,050 | 32.74 | 1,025 | 32.74 | 1,025 | 32.74 | |
| Female | 4,212 | 67.26 | 2,106 | 67.26 | 2,106 | 67.26 | |
| **Age (yrs)** | 74.26 ± 19.41 | | 74.04 ± 18.86 | | 74.47 ± 19.95 | | 0.381 |
| **Age groups (yrs)** | | | | | | | 0.961 |
| 18–49 | 538 | 8.59 | 270 | 8.62 | 268 | 8.56 | |
| 50–59 | 1,268 | 20.25 | 638 | 20.38 | 630 | 20.12 | |
| ≧ 60 | 4,456 | 71.16 | 2,223 | 71.00 | 2,233 | 71.32 | |
| **Catastrophic illness** | | | | | | | 0.926 |
| Without | 4,927 | 78.68 | 2,462 | 78.63 | 2,465 | 78.73 | |
| With | 1,335 | 21.32 | 669 | 21.37 | 666 | 21.27 | |
| **CCI_R** | 1.04 ± 1.16 | | 1.05 ± 1.17 | | 1.03 ± 1.15 | | 0.495 |
| **Level of care** | | | | | | | < 0.001 |
| Medical center | 2,668 | 42.61 | 1,597 | 51.01 | 1,071 | 34.21 | |
| Regional hospital | 2,161 | 34.51 | 1,013 | 32.35 | 1,148 | 36.67 | |
| Local hospital | 1,433 | 22.88 | 521 | 16.64 | 912 | 29.13 | |

P: Chi-square / Fisher exact test on category variables and t-test on continue variables.

including the type of TCM treatment such as herbal prescriptions, acupuncture, TCM traumatology, or a combination of herbal prescriptions. The TCM group had lower risks of fracture (adjusted HR = 0.467, 95%CI = 0.225–0.680, P<0.001), hospitalization (adjusted HR = 0.556, 95%CI = 0.330–0.751, P<0.001), and mortality (adjusted HR = 0.704, 95%CI = 0.476–0.923, P = 0.013). Information regarding the definitions of herbal formulas is shown in S1 Table.

## Discussions

This study used NHIRD data to investigate the effect of TCM on the fracture surgery, hospitalization, and all-cause mortality rates of patients with T2DOP. Our results showed that TCM treatment is a therapeutic option that can help reduce harm among patients with T2DOP. On Cox regression, factors affecting fracture surgery, hospitalization, and all-cause mortality rates included male sex, age > 50 years, and higher disease severity. However, after receiving the TCM treatment, regardless of sex, age, and disease severity, the above-mentioned risk of fracture surgery, hospitalization, and all-cause mortality showed a statistically significant reduction on the Kaplan-Meier analysis (log-rank, P<0.001).

Both diabetes and osteoporosis are affected by aging and often coexist [8]. Studies have pointed out that patients with diabetes have an increased risk of fractures. The longer the

**Table 3. Factors of prognosis by using Cox regression.**

| Prognosis | Fracture surgery | | | | Inpatient | | | | All-caused mortality | | | |
|---|---|---|---|---|---|---|---|---|---|---|---|---|
| Variables | Adjusted HR | 95% CI | 95% CI | P | Adjusted HR | 95% CI | 95% CI | P | Adjusted HR | 95% CI | 95% CI | P |
| TCM | | | | | | | | | | | | |
| Without | Reference | | | | Reference | | | | Reference | | | |
| With | 0.467 | 0.225 | 0.680 | < 0.001 | 0.556 | 0.330 | 0.751 | < 0.001 | 0.704 | 0.476 | 0.923 | 0.012 |
| Gender | | | | | | | | | | | | |
| Male | 1.239 | 0.931 | 1.625 | 0.067 | 1.301 | 1.026 | 1.510 | 0.033 | 1.333 | 0.975 | 1.816 | 0.057 |
| Female | Reference | | | | Reference | | | | Reference | | | |
| Age groups (yrs) | | | | | | | | | | | | |
| 18–49 | Reference | | | | Reference | | | | Reference | | | |
| 50–59 | 1.666 | 1.112 | 2.304 | < 0.001 | 1.276 | 0.960 | 1.556 | 0.059 | 1.454 | 1.023 | 2.026 | 0.037 |
| ≧ 60 | 1.801 | 1.144 | 2.410 | < 0.001 | 1.878 | 1.298 | 2.596 | < 0.001 | 1.880 | 1.346 | 2.455 | < 0.001 |
| Catastrophic illness | | | | | | | | | | | | |
| Without | Reference | | | | Reference | | | | Reference | | | |
| With | 1.898 | 1.456 | 2.591 | < 0.001 | 1.875 | 1.161 | 2.510 | < 0.001 | 2.015 | 1.465 | 2.810 | < 0.001 |
| CCI_R | 1.331 | 1.193 | 1.618 | < 0.001 | 1.186 | 1.005 | 1.350 | 0.047 | 1.190 | 1.087 | 1.324 | 0.007 |
| Level of care | | | | | | | | | | | | |
| Medical Center | 2.336 | 1.842 | 2.930 | < 0.001 | 2.179 | 1.481 | 2.897 | < 0.001 | 2.250 | 1.605 | 2.921 | < 0.001 |
| Regional | 1.889 | 1.420 | 2.205 | < 0.001 | 2.053 | 1.450 | 2.846 | < 0.001 | 1.765 | 1.348 | 2.327 | < 0.001 |
| Local | Reference | | | | Reference | | | | Reference | | | |

Adjusted HR = Adjusted hazard ratio: Adjusted variables listed in the table, CI = confidence interval.

Global test, P = 0.787 (Fracture surgery), 0.649 (Inpatient), and 0.895 (All-caused mortality).

**Table 4. Factors of fracture surgery, inpatient, all-caused mortality stratified by variables listed in the table by using Cox regression.**

| TCM | Fracture surgery With vs.Without (Reference) | | | | Inpatient With vs.Without (Reference) | | | | All-caused mortality With vs.Without (Reference) | | | |
|---|---|---|---|---|---|---|---|---|---|---|---|---|
| Stratified | Adjusted HR | 95% CI | 95% CI | P | Adjusted HR | 95% CI | 95% CI | P | Adjusted HR | 95% CI | 95% CI | P |
| Total | 0.467 | 0.225 | 0.680 | < 0.001 | 0.556 | 0.330 | 0.751 | < 0.001 | 0.704 | 0.476 | 0.923 | 0.012 |
| Gender | | | | | | | | | | | | |
| Male | 0.472 | 0.228 | 0.690 | < 0.001 | 0.575 | 0.342 | 0.774 | < 0.001 | 0.717 | 0.483 | 0.934 | 0.017 |
| Female | 0.457 | 0.222 | 0.677 | < 0.001 | 0.543 | 0.323 | 0.739 | < 0.001 | 0.723 | 0.472 | 0.915 | 0.007 |
| Age groups (yrs) | | | | | | | | | | | | |
| 18–49 | 0.414 | 0.195 | 0.605 | < 0.001 | 0.520 | 0.299 | 0.699 | < 0.001 | 0.684 | 0.461 | 0.895 | < 0.001 |
| 50–49 | 0.433 | 0.206 | 0.634 | < 0.001 | 0.543 | 0.324 | 0.729 | < 0.001 | 0.700 | 0.473 | 0.914 | 0.007 |
| ≧ 60 | 0.479 | 0.233 | 0.701 | < 0.001 | 0.570 | 0.337 | 0.765 | < 0.001 | 0.707 | 0.482 | 0.928 | 0.014 |
| Catastrophic illness | | | | | | | | | | | | |
| Without | 0.458 | 0.215 | 0.671 | < 0.001 | 0.548 | 0.315 | 0.740 | < 0.001 | 0.694 | 0.464 | 0.914 | 0.008 |
| With | 0.487 | 0.240 | 0.712 | < 0.001 | 0.581 | 0.349 | 0.784 | < 0.001 | 0.730 | 0.494 | 0.957 | 0.029 |
| Level of care | | | | | | | | | | | | |
| Medical center | 0.487 | 0.233 | 0.704 | < 0.001 | 0.571 | 0.336 | 0.772 | < 0.001 | 0.725 | 0.485 | 0.946 | 0.023 |
| Regional hospital | 0.467 | 0.225 | 0.682 | < 0.001 | 0.559 | 0.323 | 0.748 | < 0.001 | 0.700 | 0.471 | 0.918 | 0.009 |
| Local hospital | 0.430 | 0.206 | 0.641 | < 0.001 | 0.522 | 0.298 | 0.708 | < 0.001 | 0.636 | 0.429 | 0.846 | < 0.001 |

PYs = Person-years; Adjusted HR = Adjusted Hazard ratio: Adjusted for the variables listed in Table 3.; CI = confidence interval.

**Table 5. Factors of prognosis among different TCM subgroups by using Cox regression.**

| Prognosis | TCM subgroup | Population | Events | PYs | Rate (per $10^5$ PYs) | Adjusted HR | 95% CI | 95% CI | P |
|---|---|---|---|---|---|---|---|---|---|
| Fracture surgery | Without TCM | 3,131 | 622 | 30,439.31 | 2,043.41 | Reference | | | |
| | With TCM | 3,131 | 433 | 30,014.22 | 1,442.65 | 0.467 | 0.225 | 0.680 | < 0.001 |
| | Herbal formulae only | 2,347 | 326 | 22,498.38 | 1,448.99 | 0.468 | 0.226 | 0.684 | < 0.001 |
| | Acupuncture only | 135 | 20 | 1,294.35 | 1,545.18 | 0.500 | 0.242 | 0.729 | < 0.001 |
| | TCM traumatology only | 114 | 17 | 1,092.88 | 1,555.52 | 0.504 | 0.243 | 0.734 | < 0.001 |
| | Herbal formulae + Acupuncture | 297 | 38 | 2,847.09 | 1,334.70 | 0.431 | 0.208 | 0.629 | < 0.001 |
| | Herbal formulae + TCM traumatology | 238 | 32 | 2,281.52 | 1,402.57 | 0.454 | 0.220 | 0.661 | < 0.001 |
| | Herbal formulae | 2,882 | 396 | 27,626.99 | 1,433.38 | 0.464 | 0.224 | 0.676 | < 0.001 |
| | Acupuncture | 432 | 58 | 4,141.44 | 1,400.48 | 0.453 | 0.219 | 0.660 | < 0.001 |
| | TCM traumatology | 352 | 49 | 3,374.40 | 1,452.11 | 0.469 | 0.227 | 0.685 | < 0.001 |
| Inpatient | Without TCM | 3,131 | 845 | 33,255.24 | 2,540.95 | Reference | | | |
| | With TCM | 3,131 | 624 | 32,597.21 | 1,914.27 | 0.556 | 0.330 | 0.751 | < 0.001 |
| | Herbal formulae only | 2,347 | 469 | 24,434.55 | 1,919.41 | 0.557 | 0.331 | 0.753 | < 0.001 |
| | Acupuncture only | 135 | 26 | 1,405.63 | 1,849.70 | 0.538 | 0.319 | 0.726 | < 0.001 |
| | TCM traumatology only | 114 | 22 | 1,186.87 | 1,853.61 | 0.539 | 0.320 | 0.727 | < 0.001 |
| | Herbal formulae + Acupuncture | 297 | 59 | 3,092.25 | 1,908.00 | 0.554 | 0.329 | 0.749 | < 0.001 |
| | Herbal formulae + TCM traumatology | 238 | 48 | 2,477.91 | 1,937.12 | 0.563 | 0.334 | 0.759 | < 0.001 |
| | Herbal formulae | 2,882 | 576 | 30,004.71 | 1,919.70 | 0.557 | 0.331 | 0.754 | < 0.001 |
| | Acupuncture | 432 | 85 | 4,497.88 | 1,889.78 | 0.548 | 0.325 | 0.742 | < 0.001 |
| | TCM traumatology | 352 | 70 | 3,664.78 | 1,910.07 | 0.555 | 0.329 | 0.750 | < 0.001 |
| All-caused mortality | Without TCM | 3,131 | 397 | 46,279.43 | 857.83 | Reference | | | |
| | With TCM | 3,131 | 318 | 45,131.27 | 704.61 | 0.704 | 0.476 | 0.923 | 0.013 |
| | Herbal formulae only | 2,347 | 241 | 33,831.05 | 712.36 | 0.712 | 0.481 | 0.933 | 0.017 |
| | Acupuncture only | 135 | 14 | 1,945.63 | 719.56 | 0.719 | 0.486 | 0.943 | 0.022 |
| | TCM traumatology only | 114 | 10 | 1,643.28 | 608.54 | 0.608 | 0.411 | 0.797 | < 0.001 |
| | Herbal formulae + Acupuncture | 297 | 28 | 4,281.07 | 654.04 | 0.654 | 0.442 | 0.857 | < 0.001 |
| | Herbal formulae + TCM traumatology | 238 | 25 | 3,430.24 | 728.81 | 0.729 | 0.492 | 0.954 | 0.027 |
| | Herbal formulae | 2,882 | 294 | 41,542.36 | 707.71 | 0.707 | 0.478 | 0.927 | 0.013 |
| | Acupuncture | 432 | 42 | 6,226.70 | 674.51 | 0.674 | 0.455 | 0.884 | < 0.001 |
| | TCM traumatology | 352 | 35 | 5,073.52 | 689.86 | 0.690 | 0.466 | 0.903 | 0.002 |

PYs = Person-years; Adjusted HR = Adjusted Hazard ratio: Adjusted for the variables listed in Table 3.; CI = confidence interval.

duration of diabetes, poor blood sugar control, and the presence of diabetic vascular complications, the higher the risk of fractures, especially in patients receiving hypoglycemic drugs and those with neuropathy and retinopathy which with comorbidities are at increased risk of falls and fractures [15,16]. It has also been noted that compared with type 1 diabetes that T2D is not currently included in the fracture risk assessment tool (FRAX) used to calculate the 10-year fracture risk probability, which may be related to the fact that more than half of T2D patients are overweight or obese [17], and the impact of T2D on bone fragility is complex [18], with T2D patients generally having higher bone mineral density (BMD) than non-diabetic patients [19]. Patients with T2D often suffer from retinal macular degeneration, and one study reported that TCM Mingjing Granules can help to improve the safety and efficacy of intravitreal injections of anti–vascular endothelial growth factor in treating neovascular age-related macular degeneration [20]. This can further reduce the incidence of retinal macular degeneration in diabetic patients, a symptom which can lead to accidents such as falls and fractures due

to poor vision. T2D patients with diabetic nephropathy, secondary hyperparathyroidism, and renal osteodystrophy have also been shown to be at an increased risk of fracture [21].

Despite rapid advances in the treatment of T2D, the continued increase in the incidence of T2D indicates that currently available treatments are insufficient to reduce the prevalence of diabetes [22]. Treatment of diabetes with TCM is widely used in Taiwan, and the efficacy of TCM in T2D is also recognized in China [23]. In the treatment of T2D, the research of TCM has provided a new way, and the curative effect of TCM is worthy of further research and verification [24]. A systematic review and meta-analysis of randomized controlled trials (RCT) showed that Jinlida Granules statistically and clinically reduced fasting plasma glucose, 2 hours plasma glucose and HbA1c in patients with T2D [25]. Few recent systematic review and meta-analysis of another RCT showed that the addition of the TCM in the treatment of T2D has a statistically significant effect on reducing HbA1c in Chinese adult patients with T2D [26,27].

Similar to T2D, TCM have demonstrated therapeutic potential in the treatment of osteoporosis, often in combination with western medicine [28]. More studies have shown that these TCM play an important role in the recovery of osteoporosis by regulating the immune system [29,30]. The mode of action of TCM in the treatment of osteoporosis involves multiple pathways and targets. In general, TCM functions by promoting osteogenesis and minimizing the extremely unbalanced bone turnover, exerting anti-catabolic and anabolic effects, thus improving BMD with minimal bone microarchitectural degradation [31,32]. A recent systematic review and meta-analysis of RCT illustrated the efficacy and safety of TCM in the treatment of osteoporosis [33,34]. Recent studies have shown the therapeutic potential and outlook of alternative medicine in bone conditions, which act through exerting pro-anabolic effects and anti-catabolic effects, while regulating osteoclasts and osteoblasts [31,35]. Indeed, previous investigation has shown that bone homeostasis is regulated by osteoclasts and osteoblasts together with osteocytes, bone lining cells, osteomas, and vascular endothelial cells in the bone microenvironment within the basic multicellular unit [36]. It would be informative to investigate the osteoblastic and osteoclastic effects of TCM in terms of lowering fracture, hospitalization, and mortality risks in patients with T2DOP.

The TCM Bushen-Jianpi-Huoxue (BSJPHX) decoction has been proved in animal experiments that it may prevent diabetic osteoporosis in rats through Wnt and nuclear factor-κB signaling pathways [37]. A recent study used a systematic review and meta-analysis to explore the effectiveness of TCM treatment in T2DOP [38]. It pointed out the efficacy and safety of TCM based on principle of tonifying-kidney, strengthening-spleen, and invigorating blood circulation BSJPHX has significant efficacy against T2DOP.

Through our research, patients with T2DOP who used TCM seemed to have lower fracture surgery, hospitalization, and mortality rates. The analysis identified that patients who are male, are ≥60 years of age, have catastrophic diseases, and have higher disease severity are at higher risk of fracture surgery, which shows that closer monitoring of these patients is necessary. Our research also highlights that, regardless of the patient's sex, age, and disease severity, as long as TCM is used to treat T2DOP, a lower risk of fracture surgery, hospitalization, and all-cause death will result.

This study has several limitations that should be mentioned. Studies based on medical claims datasets are usually biased towards various clinical data, since the information on confounding factors contained in claims datasets is usually limited [39]. Patients with osteoporosis may have been missed due to the retrospective design. NHIRD did not include other residual confounders such as genetic or dietary factors and height, weight, body mass index, DEXA scores and laboratory results of the levels of bone markers such as beta-C-terminal telopeptide, tartrate-resistant acid phosphatase 5b, propeptide of type I collagen, and alkaline phosphatase. The reason why we did not use Diabetes Complications Severity Index (DCSI) is because

DCSI requires biochemical index values, but the health insurance database does not have these blood drawing values [40]. In addition, the health insurance in Taiwan will not cover visits to a Chinese medicine doctor, or prescriptions for water-cooked medicine, meaning that most medical institutions will let the patient pay for the treatment at their own expense. As a result, there is no information regarding such visits in the health insurance database, meaning that the results may be underestimated. In addition, another limitation of NHIRD that cannot show the severity of diabetes or osteoporosis, but supplementary information shows that the cumulative use time of TCMs is an average of 10.76 ± 8.03 years (S2 Table), which is similar to the results of a recent study that using TCMs group follow-up time was longer than that without using TCMs group indicating that after osteoporosis is diagnosed, the use of TCMs may delay the occurrence of fractures, and the longer the use of TCMs with lower the fracture rate [41]. Furthermore, since this was a population-based study, it was not possible to elucidate the actual mechanism underlying the association between lower risks associated with T2DOP use of TCM treatments.

## Conclusion

This nationwide population-based cohort study provides longitudinal evidence that the concomitant use of TCM and T2DOP treatment is associated with lower fracture surgery, hospitalization, and mortality rates. More research is needed to fully understand the clinical implications of the potential protective effects of TCM treatment for T2DOP.

## Supporting information

**S1 Table. Abbreviation, ICD-9-CM, and definition.**
(DOCX)

**S2 Table. Factors of fracture, inpatient, all-caused mortality stratified by variables listed in the table by using Cox regression.**
(DOCX)

**S3 Table.** 1. Years of follow-up. 2. Years to prognosis.
(DOCX)

## Acknowledgments

We appreciate the Health and Welfare Data Science Center, Ministry of Health and Welfare (HWDC, MOHW), Taiwan, for providing the National Health Insurance Reseach Database (NHIRD) and the Teh-Tzer Human Medical Research in Taiwan.

## Author Contributions

**Conceptualization:** Chien-Jung Lin, Sheng-Chiang Su, Feng-Chih Kuo, Jhih-Syuan Liu, Peng-Fei Li, Chia-Luen Huang, Li-Ju Ho, Chun-Yung Chang, Ming-Shiun Lin, Chih-Ping Lin, An-Che Cheng, Chien-Hsing Lee, Chang-Hsun Hsieh, Yi-Jen Hung, Hsin-Ya Liu.

**Formal analysis:** Chi-Hsiang Chung.

**Funding acquisition:** Chieh-Hua Lu, Wu-Chien Chien.

**Writing – original draft:** Yi-Chen Liu.

**Writing – review & editing:** Chieh-Hua Lu, Wu-Chien Chien.

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
