## [Decision Letter · Decision Letter 0]

27 Sep 2023

PONE-D-23-22181Association with Lowering Fracture, Inpatient and Mortality Risk in Patients of Diabetes with Osteoporosis Combined Traditional Chinese Medicine TherapyPLOS ONE

Dear Dr. Lu,

Thank you for submitting your manuscript to PLOS ONE. After careful consideration, we feel that it has merit but does not fully meet PLOS ONE’s publication criteria as it currently stands. Therefore, we invite you to submit a revised version of the manuscript that addresses the points raised during the review process.

We look forward to receiving your revised manuscript.

Kind regards,

Tsung-Tai Chen

Academic Editor

PLOS ONE

Additional Editor Comments:

Comments from PLOS Editorial Office: We note that one or more reviewers has recommended that you cite specific previously published works. As always, we recommend that you please review and evaluate the requested works to determine whether they are relevant and should be cited. It is not a requirement to cite these works. 

We have now received four reviews of your manuscript. The reviewers' comments are located at the end of this letter.

Upon reading these comments and your paper, we think that a major revision, perhaps involving new analyses, might lead to a publishable paper. Of course, we cannot promise a favorable decision to publish at this point.

Reviewers' comments:

Reviewer's Responses to Questions

**Comments to the Author**

1. Is the manuscript technically sound, and do the data support the conclusions?

Reviewer #1: Yes

Reviewer #2: Yes

Reviewer #3: No

Reviewer #4: Yes

2. Has the statistical analysis been performed appropriately and rigorously? 

Reviewer #1: Yes

Reviewer #2: No

Reviewer #3: No

Reviewer #4: Yes

3. Have the authors made all data underlying the findings in their manuscript fully available?

Reviewer #1: Yes

Reviewer #2: No

Reviewer #3: No

Reviewer #4: Yes

4. Is the manuscript presented in an intelligible fashion and written in standard English?

Reviewer #1: Yes

Reviewer #2: No

Reviewer #3: No

Reviewer #4: Yes

5. Review Comments to the Author

Reviewer #1: The causes of fracture are multiple factorial. Thus, the TCM treatment reduce fracture rate may through other route. For example, improve eye function thus reduce falling events, thus reduce fracture rate. I hope the authors can have more works about this.

Reviewer #2: This study utilized a nationwide health database to analyze the use of traditional Chinese medicine (TCM) among diabetic patients with or without osteoporosis and its impact on health outcomes. This is a highly innovative research approach.

I have the following major comments for the authors:

1.In the Materials and Methods section, the author did not provide clear definitions for diabetes, osteoporosis, and health outcomes, especially considering that a claims database was used as the data source.

2.There is a lack of specific drug classification explanation for the use of TCM scientifically processed (Powdered)

3.The definition of censoring is not explicitly explained.

4.For research results not provided Incidence rates for the main outcomes.

5.There is no discussion on the selection bias and its possible impact on research results.

6.Please ensure that references cited in the text are consistent."

Reviewer #3: PONE-D-23-22181: statistical review

SUMMARY.

This is a retrospective study to test whether traditional Chinese therapy reduces the risk of fracture surgery, hospitalization and total mortality in a sample of diabetic patients with osteoporosis. The core statistical analysis relies on a battery of Cox regressions models, estimated from two propensity-score matched samples. Beside the problematic intelligibility of the paper (see specific point no. 1), there are some major issues that should be addressed.

MAJOR ISSUES

1. Propensity-score matched samples can be useful to alleviate the possible bias of retrospective studies like this one. However, the matching is successful when there are not significant differences between the groups of interest (in this case the groups with and without TCM treatments). This is unfortunately not the case in this study (see table 1) because there are relevant differences in level of care and catastrophic illness. The authors should discuss why these differences are not expected to bias the results.

2. The treatment of interest is traditional Chinese medicine, but nothing is said about this treatment. Is there a specific TCM treatment for diabetic patients with osteoporosis? How long have these patients been treated by TCM?

3. The study relies on very few covariates and quite high is the risk that unobserved confounding factors could bias the analysis. I assume that the authors did not include further covariates simply because they are not available in the dataset. Undesired effects of unobserved confounders could be alleviated by introducing random effects in the Cox regression model, a method that is available in SPSS, the software used by the authors.

SPECIFIC POINTS.

1. English. The paper has several grammar and language issues, which need to be addressed. Even the title does not seem grammatically correct to me. A title like "The role of traditional Chinese medicine on fracture surgery, hospitalization and total mortality risks in diabetic patients with osteoporosis" would sound better.

2. The Cox regression model relies on the hypothesis of proportional hazards. Could the authors provide evidence that such assumption is fulfilled by the data?

Reviewer #4: The manuscript entitled “Association with Lowering Fracture, Inpatient and Mortality Risk in Patients of Diabetes with Osteoporosis Combined Traditional Chinese Medicine Therapy” is presented. The study is potentially interesting. The methods employed for analyses were reasonable and the conclusions were justified.

There are some potential issues.

Tabels 1 and table 4, Do you have Dexa scores for Patients of Diabetes?

It was mentioned that TCM has effects on osteoporosis. Recent studies have found Therapeutic Potential and Outlook of Alternative Medicine in bone conditions via pro-anabolic effects and anti-catabolic effects and the regulation of osteoclasts and osteoblasts (for example PMID: 28325144, PMID: 31824310). It would be relevant to discuss the osteoblastic effects and osteoclastic effects of TCM and how this might affect the precited association with Lowering Fracture, Inpatient and Mortality Risk in Patients of Diabetes with Osteoporosis.

Can you include serum biomarkers of bone remodelling in these analyses? such as measurement data of b-CTX, TRAP-5b , P1NP, ALP? In this study, it was mentioned that TCM has multiple effects. Many studies have found (for example PMID: 22465238 ) bone homeostasis is regulated by osteoclasts and osteoblasts together with osteocytes, bone lining cells, osteomacs, and vascular endothelial cells in the bone microenvironment within the basic multicellular unit (BMU). It would be informative to discuss the effect of diabetes on these cell types.

6. PLOS authors have the option to publish the peer review history of their article (what does this mean?). If published, this will include your full peer review and any attached files.

Reviewer #1: **Yes: **Chen-Kun Liaw, Associate Professor, Taipei Medical University

Reviewer #2: No

Reviewer #3: No

Reviewer #4: No

---

## [Author Response · Author response to Decision Letter 0]

10 Nov 2023

Dear review committee

Thank you very much for your valuable opinions. We also responded in detail bit by bit. We hope that the members can see our intentions. We also thank the members again for their suggestions to make this article more perfect.

Sincerely yours,

Wu-Chien Chien, PhD.

---

## [Decision Letter · Decision Letter 1]

12 Dec 2023

PONE-D-23-22181R1The role of traditional Chinese medicine on fracture surgery, hospitalization, and total mortality risks in diabetic patients with osteoporosisPLOS ONE

Dear Dr. Lu,

Thank you for submitting your manuscript to PLOS ONE. After careful consideration, we feel that it has merit but does not fully meet PLOS ONE’s publication criteria as it currently stands. Therefore, we invite you to submit a revised version of the manuscript that addresses the points raised during the review process. Please revise your manuscript according to the referees’ comments, especially for methodology

We look forward to receiving your revised manuscript.

Kind regards,

Tsung-Tai Chen

Academic Editor

PLOS ONE

Reviewers' comments:

Reviewer's Responses to Questions

**Comments to the Author**

1. If the authors have adequately addressed your comments raised in a previous round of review and you feel that this manuscript is now acceptable for publication, you may indicate that here to bypass the “Comments to the Author” section, enter your conflict of interest statement in the “Confidential to Editor” section, and submit your "Accept" recommendation.

Reviewer #2: (No Response)

Reviewer #3: (No Response)

Reviewer #4: All comments have been addressed

2. Is the manuscript technically sound, and do the data support the conclusions?

Reviewer #2: Yes

Reviewer #3: No

Reviewer #4: Yes

3. Has the statistical analysis been performed appropriately and rigorously? 

Reviewer #2: Yes

Reviewer #3: No

Reviewer #4: N/A

4. Have the authors made all data underlying the findings in their manuscript fully available?

Reviewer #2: Yes

Reviewer #3: No

Reviewer #4: Yes

5. Is the manuscript presented in an intelligible fashion and written in standard English?

Reviewer #2: Yes

Reviewer #3: No

Reviewer #4: Yes

6. Review Comments to the Author

Reviewer #2: Thank you for the author's response to the review comments. The author did not conduct an analysis regarding the correlation between severity of diabetes, osteoporosis, and the cumulative amount of traditional Chinese medicine usage. It is suggested that the author discuss the impact of diabetes, osteoporosis severity and indication bias on the study results to enhance this better of this manuscript.

Reviewer #3: PONE-D-23-22181R1: statistical review

SUMMARY. In my first review of this paper, I have raised 3 major issues and 2 specific points. Major issues no. 1 and 2 and specific issue no. 1 have been adequately addressed. However, there is problem with the remaining issues. See below.

1. I previously raised the following concern: "The study relies on very few covariates and quite high is the risk that unobserved

confounding factors could bias the analysis. I assume that the authors did not

include further covariates simply because they are not available in the dataset.

Undesired effects of unobserved confounders could be alleviated by introducing

random effects in the Cox regression model, a method that is available in SPSS,

the software used by the authors."

In the revised manuscript, the authors declare that the statistical analysis has been revised by including random effects in the model. However, the results in Table 3 and 4 are the same as in the previous version. Table 5 seems also to have been done without including random effects. I would like to see the new tables that have been obtained by a mixed effect Cox regression.

2. I previously raised some concerns about the English. Although the authors declare that the English has been revised, I still see mistakes.

Reviewer #4: This is a revised paper. The authors have addressed questions and the paper has been improved. It is acceptable for publication.

---

## [Author Response · Author response to Decision Letter 1]

9 Jan 2024

Reviewer #2: Thank you for the author's response to the review comments. The author did not conduct an analysis regarding the correlation between severity of diabetes, osteoporosis, and the cumulative amount of traditional Chinese medicine usage. It is suggested that the author discuss the impact of diabetes, osteoporosis severity and indication bias on the study results to enhance this better of this manuscript.

Response:

Thank you for your suggestions.

We describe in the discussion with blue words in Line 344-351, Page 20-21.

Thank you again for pointing these out.

Reviewer #3: PONE-D-23-22181R1: statistical review

SUMMARY. In my first review of this paper, I have raised 3 major issues and 2 specific points. Major issues no. 1 and 2 and specific issue no. 1 have been adequately addressed. However, there is problem with the remaining issues. See below.

1. I previously raised the following concern: "The study relies on very few covariates and quite high is the risk that unobserved confounding factors could bias the analysis. I assume that the authors did not include further covariates simply because they are not available in the dataset. Undesired effects of unobserved confounders could be alleviated by introducing random effects in the Cox regression model, a method that is available in SPSS, the software used by the authors." In the revised manuscript, the authors declare that the statistical analysis has been revised by including random effects in the model. However, the results in Table 3 and 4 are the same as in the previous version. Table 5 seems also to have been done without including random effects. I would like to see the new tables that have been obtained by a mixed effect Cox regression.

Response:

Thank you for your suggestions.

We are sorry that the data was not updated last time. We have corrected it, as shown in red letters in the attached Table.

The parts where the correction content is modified in abstracts and results are also displayed in blue.

Thank you again for pointing these out.

2. I previously raised some concerns about the English. Although the authors declare that the English has been revised, I still see mistakes.

Response:

Thank you for your suggestions.

The files for tracking revisions include corrections related to English revisions in the article.

Thank you again for pointing these out.

Reviewer #4: This is a revised paper. The authors have addressed questions and the paper has been improved. It is acceptable for publication.

Response:

Thank you for your previous advice.

Thank you for the very kind encouragement

---

## [Decision Letter · Decision Letter 2]

2 Feb 2024

PONE-D-23-22181R2The role of traditional Chinese medicine on fracture surgery, hospitalization, and total mortality risks in diabetic patients with osteoporosisPLOS ONE

Dear Dr. Lu,

Thank you for submitting your manuscript to PLOS ONE. After careful consideration, we feel that it has merit but does not fully meet PLOS ONE’s publication criteria as it currently stands. Therefore, we invite you to submit a revised version of the manuscript that addresses the points raised during the review process.

1 The distribution between the experimental and control groups differs, notably in the variable "catastrophic illness." Such difference may significantly impact outcomes, urging consideration of this variable during matching procedures.

2 It is recommended to provide documentation verifying the edits made to the manuscript after revision. Some English errors persist, including for example, inaccurate descriptions in tables such as "hospital center”

3 

(1) Inconsistencies exist regarding the terms "comorbidity" and "severity" throughout the article. Confirm that the Charlson Comorbidity Index (CCI) represents comorbidity, while catastrophic illness denotes severity. 

(2) Including literature to support the revised CCI.

(3) There were many studies utilizing database-derived measure like the Diabetes Complications Severity Index (DCSI) to demonstrate severity for patients with diabetes. Please amend the "Limitations" section to reflect this aspect accurately.

4 Please explain the results of Schoenfeld's global test and what the assumptions required for proportional hazard analysis the test satisfies. 

We look forward to receiving your revised manuscript.

Kind regards,

Tsung-Tai Chen

Academic Editor

PLOS ONE

Journal Requirements:

Reviewers' comments:

Reviewer's Responses to Questions

**Comments to the Author**

Reviewer #2: All comments have been addressed

Reviewer #3: All comments have been addressed

2. Is the manuscript technically sound, and do the data support the conclusions?

Reviewer #2: Yes

Reviewer #3: (No Response)

3. Has the statistical analysis been performed appropriately and rigorously? 

Reviewer #2: Yes

Reviewer #3: (No Response)

4. Have the authors made all data underlying the findings in their manuscript fully available?

Reviewer #2: Yes

Reviewer #3: (No Response)

5. Is the manuscript presented in an intelligible fashion and written in standard English?

Reviewer #2: Yes

Reviewer #3: (No Response)

6. Review Comments to the Author

Reviewer #2: Reviewer has undergone a thorough review, and on its significant translational value. reviewer appreciate the potential impact of research and believe it deserves acceptance. this manuscript aligns with journal's standards and contributes meaningfully to the field.

Reviewer #3: (No Response)

7. PLOS authors have the option to publish the peer review history of their article (what does this mean?). If published, this will include your full peer review and any attached files.

Reviewer #2: No

Reviewer #3: No

---

## [Author Response · Author response to Decision Letter 2]

28 Feb 2024

Association with Lowering Fracture, Inpatient and Mortality Risk in Patients of Diabetes with Osteoporosis Combined Traditional Chinese Medicine Therapy

RE:PLOS ONE Decision: Revision required [PONE-D-23-22181-R2]

Lu et al. 

1. The distribution between the experimental and control groups differs, notably in the variable "catastrophic illness." Such difference may significantly impact outcomes, urging consideration of this variable during matching procedures.

Response:

Thank you for your suggestions.

Following the reviewer's suggestions, we make the distributions similar between the experimental and control groups at the beginning of the analysis and displayed the faithful description based on the new figures and tables in green fonts in the article.

Thank you again for pointing these out.

2. It is recommended to provide documentation verifying the edits made to the manuscript after revision. Some English errors persist, including for example, inaccurate descriptions in tables such as "hospital center.”

Response:

Thank you for your suggestions.

We have corrected it, as shown in green fonts in the attached Table.

Thank you again for pointing these out.

3. Inconsistencies exist regarding the terms "comorbidity" and "severity" throughout the article. Confirm that the Charlson Comorbidity Index (CCI) represents comorbidity, while catastrophic illness denotes severity. Including literature to support the revised CCI. There were many studies utilizing database-derived measure like the Diabetes Complications Severity Index (DCSI) to demonstrate severity for patients with diabetes. Please amend the "Limitations" section to reflect this aspect accurately.

Response:

Thank you for your suggestions.

1. We have corrected it, as shown in green fonts in Line 215, 222 Page 13 and Line 230 Page 14.

2. The reason why we did not use Diabetes Complications Severity Index (DCSI) is because DCSI requires biochemical index values, but the health insurance database does not have them, and it has been written into the limitation in Line 340-343, Page 20 with reference 38.

Thank you again for pointing these out.

4. Please explain the results of Schoenfeld's global test and what the assumptions required for proportional hazard analysis the test satisfies. 

Response:

Thank you for your suggestions.

As a result of Schoenfeld's global test, if P > 0.05, it means that the Proportional Hazard Assumption is not violated. In other words, the Cox proportional hazards model can be used. Cox proportional hazards model assumes that the hazards of the different strata formed by the levels of the covariates are proportional.

Reference as J.Natn.Sci.Foundation Sri Lanka 2009 37(1):41-5. “Use of Schoenfeld’s global test to test the proportional hazards assumption in the Cox proportional hazards model: an application to a clinical study”

Thank you again for pointing these out.

---

## [Editor Report · Decision Letter 3]

6 Mar 2024

PONE-D-23-22181R3The role of traditional Chinese medicine on fracture surgery, hospitalization, and total mortality risks in diabetic patients with osteoporosisPLOS ONE

Dear Dr. Lu,

Thank you for submitting your manuscript to PLOS ONE. After careful consideration, we feel that it has merit but does not fully meet PLOS ONE’s publication criteria as it currently stands. Therefore, we invite you to submit a revised version of the manuscript that addresses the points raised during the review process.

We look forward to receiving your revised manuscript.

Kind regards,

Tsung-Tai Chen

Academic Editor

PLOS ONE

Journal Requirements:

**Additional Editor Comments:**

1 There may be more than one instance of grammatical issues (the reviewer just proposed an example). The editing certification is on November 4, 2023, and the manuscript has undergone several revisions after November 4, 2023. It is advisable to provide newest editing certification

2 Please include this description of Schoenfeld's global test and its underlying assumptions, inclusive of relevant literature, in the manuscript.

---

## [Author Response · Author response to Decision Letter 3]

26 Mar 2024

Journal Requirements:

Response:

Thank you for your suggestions.

We have followed the journal requirements to check all reference lists to ensure they are complete and correct.

Additional Editor Comments:

1. There may be more than one instance of grammatical issues (the reviewer just proposed an example). The editing certification is on November 4, 2023, and the manuscript has undergone several revisions after November 4, 2023. It is advisable to provide newest editing certification.

Response:

Thank you for your suggestions.

During the submission period, the manuscript has been revised after many reviewers' suggestions. We have also sent the last revised version to the English editing unit for revision, and provided the latest editorial certification. The date of completion of editing certification is Mar 14, 2024.

Thank you again for pointing these out.

2 Please include this description of Schoenfeld's global test and its underlying assumptions, inclusive of relevant literature, in the manuscript.

Response:

Thank you for your suggestions.

We have mentioned it, as shown in orange color words in Line 156-160 Page 9 in Statistical Analysis of Materials and Methods section.

Cox proportional hazards model assumes that the hazards of the different strata formed by the levels of the covariates are proportional. Schoenfeld's global test is used to test the residuals of Cox proportional model. If P > 0.05, it means that the proportional hazard assumption is not violated; in other words, the Cox proportional hazards model can be used.

With references 13 and 14.

13. SCHOENFELD D. Chi-squared goodness-of-fit tests for the proportional hazards regression model. Biometrika. 1980;67(1):145-53. doi: 10.1093/biomet/67.1.145.

14. Abeysekera WWM, Sooriyarachchi MR. Use of Schoenfeld’s global test to test the proportional hazards assumption in the Cox proportional hazards model: an application to a clinical study. Journal of the National Science Foundation of Sri Lanka. 2009. doi: 10.4038/jnsfsr.v37i1.456.

Thank you again for pointing these out.

---

## [Editor Report · Decision Letter 4]

2 Apr 2024

The role of traditional Chinese medicine on fracture surgery, hospitalization, and total mortality risks in diabetic patients with osteoporosis

PONE-D-23-22181R4

Dear Dr. Lu,

We’re pleased to inform you that your manuscript has been judged scientifically suitable for publication and will be formally accepted for publication once it meets all outstanding technical requirements.

Kind regards,

Tsung-Tai Chen

Academic Editor

PLOS ONE